# Transcriptome Adaptation of the Ovine Mammary Gland to Dietary Supplementation of Extruded Linseed

**DOI:** 10.3390/ani11092707

**Published:** 2021-09-16

**Authors:** Giuseppe Conte, Tommaso Giordani, Alberto Vangelisti, Andrea Serra, Mariano Pauselli, Andrea Cavallini, Marcello Mele

**Affiliations:** 1Department of Agriculture, Food and Environment, University of Pisa, Via del Borghetto, 80, 56124 Pisa, Italy; tommaso.giordani@unipi.it (T.G.); albertovangelisti@libero.it (A.V.); andrea.serra@unipi.it (A.S.); andrea.cavallini@unipi.it (A.C.); marcello.mele@unipi.it (M.M.); 2Research Center of Nutraceutical and Food for Health, University of Pisa, Via del Borghetto, 80, 56124 Pisa, Italy; 3Department of Agriculture, Food and Environmental Sciences, University of Perugia, Borgo XX Giugno, 74, 06121 Perugia, Italy; mariano.pauselli@unipg.it

**Keywords:** linseed, transcriptome, RNA-Seq, mammary gland, sheep

## Abstract

**Simple Summary:**

Milk fatty acid composition and gene expression in the mammary gland were evaluated in dairy ewes supplemented with linseed. The linseed supplementation improves the health and nutrition quality properties of dairy milk, besides affecting the expression of gene networks related to energy balance, RNA binding, circadian rhythm, and nutrient metabolism. This molecular knowledge on the physiology of the mammary gland might provide the basis for more detailed functional studies for future research.

**Abstract:**

Several dietary strategies were adopted to reduce saturated fatty acids and increase beneficial fatty acids (FA) for human health. Few studies are available about the pathways/genes involved in these processes. Illumina RNA-sequencing was used to investigate changes in the ovine mammary gland transcriptome following supplemental feeding with 20% extruded linseed. Comisana ewes in mid-lactation were fed a control diet for 28 days (control period) followed by supplementation with 20% DM of linseed panel for 28 days (treatment period). Milk production was decreased by 30.46% with linseed supplementation. Moreover, a significant reduction in fat, protein and lactose secretion was also observed. Several unsaturated FAs were increased while short and medium chain saturated FAs were decreased by linseed treatment. Around four thousand (1795 up- and 2133 down-regulated) genes were significantly differentially regulated by linseed supplementation. The main pathways affected by linseed supplementation were those involved in the energy balance of the mammary gland. Principally, the mammary gland of fed linseed sheep showed a reduced abundance of transcripts related to the synthesis of lipids and carbohydrates and oxidative phosphorylation. Our study suggests that the observed decrease in milk saturated FA was correlated to down-regulation of genes in the lipid synthesis and lipid metabolism pathways.

## 1. Introduction

Ruminant milk provides key nutrients for humans and, in the last decade, many research activities aimed to improve the content of beneficial components in milk. Numerous studies on milk fat composition have been conducted to find the best strategies to improve the concentration of milk fatty acids (FAs) with potential protective roles against several human diseases, such as increased concentration of conjugated linoleic acid (CLA) with known anti-obesity, anti-carcinogenic, anti-atherogenic and immunomodulatory effects [1]. The strategies considered to achieve this goal may be summarized in two main fields: animal feeding and genetic improvement [1]. Feeding approaches include the use of supplements rich in unsaturated FAs (i.e., soybean oil, corn oil, safflower oil, linseed oil, canola oil, marine algae, and fish oil) in the diet of dairy animals [2,3,4]. In dairy cows, these strategies led to a modification in mammary gland metabolism, especially the pathway involved in lipid metabolism [5]. De novo synthesis of FAs is usually decreased (milk fat depression (MFD)), favoring an increase in unsaturated FAs in milk fat. The MFD mechanism involves alterations in rumen biohydrogenation of dietary polyunsaturated fatty acids to produce unique FA intermediates, such as trans-10, cis-12 CLA, which inhibits milk fat synthesis [5]. Contrariwise to cows, ewes are less sensitive to the effect of plant oil-containing supplementation with effects depending on the dose [3], but they may suffer milk fat depression when their diets are supplemented with marine lipids [6].

Linseed (both as oil and whole seed) is the fat source supplement in ruminant’s diet with a high content of α-linolenic acid (ALA) (about 50 to 60% of total fat). Hence, linseed supplementation in ruminant’s diet was considered an effective way to improve milk fat quality [2,5,7]. Dietary supplementation of ruminants’ diets with different forms of linseed, has been shown to increase ALA, CLA, and other omega-3 FA content of bovine [8] and ovine milk [9]. α-linolenic acid is a polyunsaturated fatty acid (PUFA) that results in different intermediates of ruminal biohydrogenation affecting the pathways of lipid metabolism [10]. The composition of biohydrogenation intermediates depends on the composition of the diet as demonstrated by several studies, which revealed the formation of different C18:1, C18:2, C18:3 and CLA isomers [10].

Diet-induced changes in mammary lipogenesis and fat secretion are determined by a coordinated modification in mRNA abundance for key enzymes involved in the biochemical pathways of fat synthesis. A more complete identification of cellular mechanisms may offer broader opportunities for application and understanding of the regulation of lipid metabolism. In this sense, beside studying diet-induced changes in mammary lipogenesis and fat secretion, recently, dairy ewes’ molecular mechanisms have been also investigated. For example, MFD and changes in FA composition caused by fish oil or trans-10, cis-12- CLA supplement were related to changes in transcript abundance of some candidate lipogenic genes and transcription factors. In particular, down-regulation of the mRNA abundance of genes involved in FA transport/uptake, de novo FA synthesis, desaturation of FA, and triglyceride synthesis were related to MFD [3,4,11]. However, these approaches based on the study of a few candidate genes provided only a partial overview of the mammary gland metabolism, lacking information about metabolic processes not related to lipid metabolism.

Thanks to the -omic technologies, metabolic and biological modification in mammary glands has been explored in more depth. RNA-sequencing (RNA-seq) is acknowledged as a sensitive, broad-spectrum detection tool for identifying dynamic gene expression profiles underlying molecular and cellular processes [12]. The rapid development of RNA-seq has made it possible to sequence a large-scale whole transcriptome, providing a deeper knowledge of transcriptomic regulation [12]. This approach has been successfully used to study the milk/mammary tissue transcriptome, although only a few studies have evaluated the effect of PUFA on the mammary gland transcriptome [12,13,14,15].

In this study, we applied RNA-Seq to gain insights into the regulatory mechanisms underlying the nutrient effect on milk fat synthesis of ovine mammary glands, in one of the most important Italian ovine dairy breeds.

## 2. Materials and Methods

### 2.1. Experimental Design and Sampling

The animals involved in the test were raised at the animal facility of the Department of Agriculture, Food and Environmental Sciences (University of Perugia) and we obtained written informed consent to use the animals in our study from the same department, which represents the animal owner. The handling of the animals was carried out according to the guidelines of the EU Directive 2010/63/EU for animal experiments and to the Institutional Animal Care and Use Committee of the University of Perugia.

Eight multiparous Comisana ewes in mid-lactation (97 ± 12 days in milking), producing 735 ± 15 g/d of milk with a weight of 65 ± 8 kg, were fed a control diet (C diet, with no lipid supplementation). After 23 days (first experimental period), all ewes were switched to extruded linseed diet (L diet) composed of a total mixed ratio (particle size > 3 cm in length) (Appendix A) administered ad libitum with 800 g/head/day of a concentrate formulated to contain 10 g/100 g of dry matter (DM) of extruded linseed (experimental period, which lasted 28 days). All concentrates were obtained by pelleting the ingredients (diameter was 5 mm) and offered in two equal doses with rolled barley, during each milking at 7:30 a.m. and 5:30 p.m. The L diet was formulated to be isoproteic and isoenergetic with the C diet, according to the nutrient requirements of an ewe weighing 68 kg and producing 1 kg of milk at 6.5% fat. Animals had free access to water. Dry matter intake (DMI) of concentrates was registered daily and individually based on residuals.

Before the start of L diet supplementation and at the end of 28 days of treatment, individual milk was sampled for each ewe. Control and linseed diets were given to the same animals in two distinct periods, giving the opportunity to evaluate the effect of treatments on gene expression without the influence of different genetic components. This design, in which each ewe got both of the interventions in series, was chosen because it reduces variability of gene expression between ewes due to heterogeneity in genetic background and increases the statistical power to detect differential gene expression to a specified type I error rate. This approach gave us the opportunity to use only 8 animals per treatment.

Milk samples were collected on day −3 (C group) and day +28 (L group) relative to the start of the treatment period, for the determination of milk composition, FA profile and RNA extraction.

The milk samples were divided into two aliquots: about 500 mL were kept at 4 °C for the period necessary to arrive in the laboratory, then the RNA extraction was immediately carried out. On the contrary, 50 mL of milk were used for the determination of the fatty acid profile and kept at −20 °C until the moment of analysis.

### 2.2. Determination of Feed and Milk Composition and Fatty Acid Profile

Feed samples of each experimental diet were analyzed for dry matter content (ISO 6496, 1998), ash (ISO 5984, 2002), Kjeldahl N (ISO 5983, 2005), ADL, ADF and NDF [16] according to a modified AOAC methods [17].

Milk composition (fat, protein and lactose percentage) was determined by a MilkoScan FT 6000 Series mid-range infrared Fourier transform infrared-based spectrometer (Foss, Hillerod, Denmark). Total lipids from the milk were trans-esterified using a 0.5 N methanolic solution of sodium methoxide [18] after adding the internal standards (C9:0 and C19:0 methyl ester). FA methyl esters (FAME) were separated and identified using a GC-FID apparatus (GC 2000 plus, Shimadzu, Columbia, MD, USA) according to Conte et al. [19].

### 2.3. RNA Extraction

Total RNA was isolated from Milk Somatic Cells (MSC), following the procedure described by Giordani et al. [20]. Briefly, MSC were pelleted from 250 mL fresh milk at 1100 g for 10 min at 4 °C. The cell pellet was washed twice in phosphate-buffered saline solution at pH 7.2 with 0.5 mM EDTA and resuspended in 3 mL of TriZol reagent (Invitrogen, Breda, the Netherlands). RNA extraction continued following the TriZol protocol, except that all centrifugations were performed at 4 °C and precipitation in isopropanol was carried out at −80 °C for 30 min. Then, DNAse I (Roche, Basel, Switzerland) treatment was performed according to the manufacturer’s instructions, to completely remove genomic DNA contamination. Finally, RNA was purified by phenol/chloroform extraction and precipitated following standard procedures. Total RNA quality was checked using a Bioanalyzer 2100 (Agilent Technologies, Santa Clara, CA, USA).

### 2.4. RNA Sequencing

RNA-Seq libraries of MSC from control (C) and linseed fed sheep (L), with four replicates each, were generated using the TruSeq RNA-Seq Sample Prep kit according to the manufacturer’s protocol (Illumina Inc., San Diego, CA, USA). Poly-A RNAs were isolated from total RNA and chemically fragmented. First and second strand cDNA synthesis were followed by end-repair and adenosines were added to the 3′ ends. Adapters were ligated to the cDNA and 200 ± 25 bp fragments were gel purified and enriched by PCR. Overall, 8 libraries were quantified using Bioanalyzer 2100 (Agilent Technologies, Santa Clara, CA, USA) and run on the Illumina HiSeq2000 (Illumina Inc.) using version 3 reagents. Paired-read sequences, 125 nt in length, were collected.

Raw single reads (in FASTQ format) were subjected to sequence quality control using FastQC (http://www.bioinformatics.babraham.ac.uk/projects/fastqc/ (accessed on 8 January 2018)). Then, raw reads were trimmed using Trimmomatic [21], version 0.33, with the following parameters: CROP: 115 HEADCROP: 15 MINLEN: 100. After trimming, ribosomal reads were mapped against *Ovis aries* and *Bos taurus* ribosomal DNA sequences and rDNA (obtained from NCBI repository), and matching reads were removed. Finally, FastQC was used again to examine the characteristics of the libraries and to verify trimming efficiency.

Reads were aligned to the *Ovis aries* transcriptome (Oar_rambouillet), version 1.0, available at the NCBI site (https://www.ncbi.nlm.nih.gov/genome/?term=ovis%20aries (accessed on 8 January 2019)), using CLC-BIO Genomic Workbench 12.0.3 (CLC) with the following parameters: length fraction = 0.8, similarity fraction = 0.8, mismatch cost = 2, insertion/deletion cost = 3.

Raw reads were deposited on SRA under PRJNA759418 bioproject accession code.

### 2.5. Differential Expression Analysis

Raw counts of reads per gene after mapping procedure were used to establish gene expression. Gene expression level was calculated as Reads Per Kilobase per Million (RPKM) as described by Mortazavi et al. [22]. Genes with RPKM >1 in at least one library of C or L sheep were retained. Expression data were evaluated by considering RPKM values, performing pairwise comparisons between C and L fed Comisana ewes using the likelihood ratio test on edgeR [23], as suggested by the instruction manual and by Anders et al. [24]. The log fold changes between treatments were considered as significant when the weight of a sample was at least one-fold higher or lower than another, with an FDR corrected *p*-value ≤ 0.05 [25]. The complete set of DEGs is available in Appendix A.

### 2.6. Transcriptome Functional Analysis

In order to get biological insights of differentially expressed genes, both gene ontology (GO) functions and KEGG metabolic pathways were analyzed. The whole set of genes expressed during the experiment and transcripts specifically regulated by treatments were annotated by GO analysis using Blast2GO [26]; GO slim was run in order to reduce GO complexity. In addition, KEGG (Kyoto Encyclopedia of Genes and Genomes) was used for metabolic pathways analyses. Enrichment tests between the metabolic pathway of DEGs and of the whole *Ovis aries* transcriptome were performed (Fisher’s exact test, *p*-value < 0.05, no correction for multiple testing was applied) in order to explain how these genes are involved in the physiological and metabolic functions occurring in the mammary gland, with or without linseed supplementation.

### 2.7. Statistical Analysis

Data related to milk production, fat, lactose, proteins and FA composition of milk were analyzed using the following linear model, using JMP software (SAS Institute Inc., Cary, NC, USA):y_ijz_ = μ + D_i_ + A_j_ + ε_ijz_
where y_ijz_ = dependent variables; D_i_ = fixed effect of the i^th^ diet treatment (C; L); A_j_ = random effect of j^th^ ewe (8 animals), ε_ijz_ = random residual. The differences between C and L groups were significant when *p* < 0.05.

## 3. Results

### 3.1. Effect of Treatment on Milk Composition and Fatty Acid Profile

Linseed supplementation significantly reduced milk production (−30%), with a significant (*p* < 0.01) decrease in fat (−25%), protein (−52%) and lactose (−32%) secretion (Table 1).

The concentration of several saturated FAs (SFAs) (C4:0, C6:0, C8:0, C12:0, C14:0, C16:0, C17:0 and total SFA) decreased significantly (*p* < 0.001) as a consequence of linseed supplementation. On the contrary, stearic acid (C18:0) was the only SFA showing a significant (*p* < 0.001) increase (Table 1). The level of monounsaturated FAs (MUFAs) was significantly (*p* < 0.001) increased with linseed (L) diet, except for C14:1c9 and C16:1c9. The content of n6 FA decreased, while n3 FA significantly increased, mainly due to the variation in C18:2n6 and C18:3n3 concentrations for n6 and n3 FA, respectively. Regarding the content of very long chain PUFA (C > 20), a significant decrease was observed in the milk of animals fed the L diet. However, linseed supplementation favored a higher (*p* < 0.001) content of rumenic acid (C18:2c9t11), the main CLA isomer in milk fat.

### 3.2. cDNA Sequencing and Aligning on the Reference Predicted Transcriptome

High quality paired end Illumina reads of 100 bp from milk somatic cells (MSC) of the same sheep fed control (C) diet and after linseed supplementation were used. Average raw reads from mRNA libraries from MSC of control and linseed ewes were approximately 32 and 38 million, respectively (Table 2) and were subjected to RNA-seq transcriptomic analysis. After removal of the low-quality reads, the number of paired-end reads spanned from 39,530,734 to 11,941,984 (Table 2). Among control and linseed libraries, an average of 88.35% and 88.60% of reads were successfully mapped to the reference sheep transcriptome (NCBI, Oar_rambouillet_v1.0), ranging from 37,211,397 to 10,888,233 aligned reads (Table 2). Based on a FDR corrected *p*-value ≤ 0.05 and |log2 FC| ≥ 1, a total of 3928 differential expressed genes (DEGs), of which 1795 upregulated and 2133 downregulated (Appendix A), were identified in the MSC of linseed versus control ewes.

### 3.3. Analysis of Differential Expressed Genes

The whole sets of expressed genes (reads per kilobase per million reads mapped (RPKM) > 1 in at least one library) of both C and L sheep were analyzed by GO (Figure 1) and split into three major classes: biological process (BP), molecular function (MF) and cellular component (CC). Regarding BP, the most distributed GO terms were “cellular process” (GO:0009987), “metabolic process” (GO:0008152) and “cellular metabolic process” (GO:0044237). Most expressed genes in MF were represented by “Binding” (GO:0005488), “Ion binding” (GO:0043167) and “Catalytic activity” (GO:0003824). Finally, concerning CC, we retrieved “Cell” (GO:0005623), “Cell part” (GO:0044464), and “Intracellular” (GO:0005622) as the most represented GO terms (Figure 1).

Deeper analysis on GO distribution of over- (OE) and under-regulated (UE) DEGs in the mammary gland of sheep after linseed feeding showed that many terms were shared, although a higher distribution occurred for under expressed transcripts (Figure 2). Overall 237 GO categories under the class BP were assigned to the transcripts, of which the largest proportion was “Cellular process”, followed by “Metabolic process” (Figure 1). “Biosynthetic process” (GO:0009058) was the BP GO term most affected by linseed supplementation (433 and 630 over- and under-expressed gene respectively), followed by “cellular component organization” (GO:0016043) showing 383 up-regulated and 528 under-regulated transcripts (Figure 2). In addition, “cellular protein modification processes” showed a large representation with 279 and 534 respectively over and under regulated transcripts. A total of 81 GO categories under MF were assigned to the expressed transcripts (Figure 1). Considering DEGs, the largest portion of genes was represented by “ion binding” (GO:0043167; 433 OE, 652 UE) and “RNA binding” (GO:0003723; 184 OE, 269 UE) (Figure 2).

Finally, considering CC, overall 24 GO terms were retrieved, amongst these terms, genes mostly belonged to “organelle” (GO:0043226, represented by 1113 OE and 1352 UE DEGs), and “cytoplasm” (GO:0005737 with 966 and 1036 respectively activated and repressed transcripts) (Figure 2).

### 3.4. KEGG Analysis

Biological pathways associated with DEGs in ewe mammary glands were identified by KEGG [27]. A total of 1175 differentially expressed genes (602 under- and 573 over-expressed genes) were assigned to 68 KEGG pathways (Appendix A). Among these pathways, DEGs were mostly annotated to “genetic information processing” (120 over- and 128 under-expressed), “signal transduction” (158 over- and 258 under-expressed), “cellular processes” (58 over- and 64 under-expressed), “lipid metabolism” (55 over- and 35 under-expressed), “carbohydrate metabolism” (62 over- and 24 under-expressed), “glycan metabolism” (23 over- and 23 under-expressed), “vitamins and cofactors metabolism” (11 over- and 13 under-expressed) and “energy metabolism” (27 over- and 3 under-expressed) pathways (Appendix A).

KEGG enrichment analysis showed that the most significantly enriched pathways were: “Basal transcription factors” (KEGG code oas03022), “Proteasome” (KEGG code oas03050), “Spliceosome” (KEGG code oas03040), “Ribosome” (KEGG code oas03010), “RNA transport” (KEGG code oas03013), “mRNA surveillance pathway” (KEGG code oas03015), “Cytokine–cytokine receptor interaction” (KEGG code oas04060), “P53 signaling pathway” (KEGG code oas04115), “Lysosome” (KEGG code oas04142), “Tight junction” (KEGG code oas04530), “Amino sugar and nucleotide sugar metabolism” (KEGG code oas00520), “Fructose and mannose metabolism” (KEGG code oas00051), “Oxidative phosphorylation” (KEGG code oas00190), “Glycerophospholipid metabolism” (KEGG code oas00564), “Glycolysis/gluconeogenesis” (KEGG code oas00010), and “SNARE interaction in vesicular transport” (KEGG code oas04130) (Figure 3).

Moreover, we observed a metabolic adaptation of the mammary gland to the expression of genes related to pathways involved in energy balance. In particular, the L diet reduced the expression of genes involved in fat and carbohydrate synthesis and increased the expression of genes related to carbohydrate catabolism (Appendix A). As demonstrated by Appendix A, ewes fed with L diet showed an over-expression of genes involved in “Glycolysis” (ATP-dependent 6-phosphofructokinase, liver type—PFKL; Fructose-bisphosphate aldolase A—ALDOA; Alcohol dehydrogenase (NADP(+))—AKR1A1), “Pyruvate metabolism” (L-lactate dehydrogenase A chain—LDHA; D-lactate dehydrogenase—LDHD) and “TCA cycle” (isocitrate dehydrogenase (NAD) subunit gamma—IDH3G; Succinate-CoA ligase—SUCLG1; Succinate dehydrogenase—SDHB; Fumarate hydratase—FH; Malate dehydrogenase—MDH2). The same situation was observed for fat metabolism related genes, with a simultaneous increase and decrease in lipolytic and liposynthetic related transcripts, respectively. The “fatty acid degradation” pathway showed several over-expressed genes (long chain fatty-acid-CoA ligase 6—ACSL6; enoyl-CoA hydratase—ECHS1; acetyl-CoA acyltransferase 1—ACAA1; enoyl-CoA delta isomerase 1—ECI1). On the contrary, we observed an under-expression of genes involved in fatty acid biosynthesis (elongation of very long chain fatty acids protein 5 and 6—ELOVL5 and ELOVL6; steroyl CoA-desaturase—SCD).

## 4. Discussion

Illumina RNA-Seq was applied to study the biological modification of the transcriptome in the ovine mammary gland as a consequence of dietary supplementation with extruded linseed. Control (without linseed) and linseed diets were distributed to the same animals in two distinct periods, giving the opportunity to evaluate the effect of treatments on gene expression without the influence of different genetic components (related to the supplementation of the two diets in different groups). This scheme, in which each ewe receives both diets in sequence, was preferred because it decreases variability of gene expression between animals as a consequence of heterogeneity in genetic background. Moreover, this design increases the statistical power to detect differential gene expression to a specified Type I error rate. A disadvantage of this design is that the effects attributed to unsaturated FA supplementation may be partly confused with effects due to difference (of 28 days) in lactation stage. However, experimental animals were all in the same stage of lactation (mid-lactation) and milk was sampled within a short period to avoid significant effects on ewe performance (i.e., milk production and fat and protein secretion). As demonstrated by Portolano et al. [28], Comisana pluriparous sheep are characterized by a lactation curve with a long persistence and a reduced slope, especially those that give birth in autumn. This means that in mid lactation (100–150 days in milking), the productive performance and gene expression are not significantly modified, as also observed in previous studies on cows [12,13]. Since, in the present work, we considered multiparous sheep lambing in autumn, and the two treatments were done between 97 and 147 days approximately, we can consider that the lactation effect did not significantly affect the mammary metabolism and the observed differences can be attributed to the different treatments.

In our experiment, linseed supplementation led to a reduction in milk production, fat, protein and lactose secretion and to a significant modification of milk FA composition including a decrease in n6 FA and an increase in n3 FA. These results are consistent with studies on ewes fed with conjugated linoleic acid isomer [3,6,29]. In other studies, on ewes fed with fish oil or trans-10 cis-12 CLA, a similar reduction in fat secretion but no differences in milk production, FA profile, and protein and lactose secretion were observed [15]. These differences confirm that the effects on milk production and milk quality are strongly affected by basal dietary forage source and on oil supplement [29].

The sequenced samples showed a high correlation based on the expressed genes, ranging between 0.57 and 0.75. This demonstrates a strict uniformity between the subjects considered, which is an index of uniformity of the mammary metabolism.

### 4.1. Differentially Expressed Genes in Mammary Glands

The functions of the expressed genes were annotated by GO and KEGG analyses. Gene ontology is often used to standardize representation of genes and provides a set of structured and controlled vocabularies for annotating genes, gene products, and sequences [30]. Figure 1 shows that “cellular process” and “metabolic process” were the most affected GO terms by linseed supplementation among the BP class. These results are consistent with the biological characteristics and lactation status during the transition period and lactation evolution [13,14,15].

Regarding MF class, we observed many DEGs in “Ion binding” and “RNA binding” as a consequence of linseed supplementation. Previous studies showed that a large proportion of expressed genes in mammary glands of rats are related to the “RNA-binding” molecular functions [31]. Other genes belonging to the MF class such as cytokine binding and cofactor binding were differentially expressed in ewes fed with trans-10 cis-12 CLA [15]. Thereby, the results of the present study confirmed that many different “binding” activities may play an important role in the physiological function of ewe mammary glands.

Finally, the terms most affected by linseed supplementation among CC class were “Organelle”, “Cytoplasm” and “Nucleoplasm” suggesting a deep remodelling at the cellular level. In addition, “mitochondrion” is one of the most represented GO terms among the cellular component for the over-expressed genes, while it is the lowest for under-expressed transcripts (Figure 2). This result suggests a regulatory effect of linseed supplementation on the physiological energy balance of mammary glands. As reported in a previous work, cytoplasm and mitochondrial activity regulate the energy metabolism, by the control of lipid and carbohydrate pathways in the ruminant mammary gland [32].

#### Effect of Linseed Supplementation on Energy Balance and Protein Pathways

KEGG analysis predicted that the expressed genes were involved in 68 pathways, of which “Carbohydrate metabolism”, “Lipid metabolism” and “Signal transduction” were amongst the most enriched (Appendix A).

Diet change induced a metabolic adaptation, highly regulated at the transcriptional level, as shown by the significant modification in the basal transcriptional factor activity, for example by the high number of DEGs observed for the KEGG category “genetic information processing”. Several transcription factors showed different expression, for example, *TFIID*, *TFIIF* and *TFIIH*, which represent important protein complexes, having roles in transcription of various protein-coding genes and DNA nucleotide excision repair pathways (Appendix A) [33].

Concerning metabolic adaptation, we ascertained a modification of the expression of some genes controlling components of circadian rhythm that are involved in the control of locomotor activity, metabolism, and behavior [34]. *Protein phosphatase 1 catalytic* (*PPP1CC*) gene was down-regulated with L diet. This gene encodes protein phosphatase 1 (PP1), which is involved in the regulation of glycogen metabolism and protein synthesis [35]. Moreover, it is involved in the regulation of the speed and rhythmicity of PER1 and PER2 phosphorylation, two proteins that play a central role in setting the speed (period) and phase of the circadian clock [34], and that connect the circadian clock to energy metabolism [35], by the regulation of glycogen metabolism and protein synthesis [36]. Wang et al. [34] showed that PER1 and PER2 genes are expressed during transition from pregnancy into lactation in cows’ mammary glands.

Energy balance (EB) is an important parameter in dairy ruminants, defined as the difference between energy intake from feed and energy required for body maintenance, gestation and milk production, which represent an energetically costly process [32]. In ruminants, the activation of carbohydrate metabolism and subsequent ATP synthesis and glucose metabolism is crucial to satisfy the ATP requirement for milk production and milk protein synthesis [32].

In our experiment, L diet induced a sensible modification in the regulation of genes directly involved in the energy metabolism (both carbohydrate and lipid metabolism). Results indicate that nutrient metabolism adaptation in the mammary gland occur as consequence of the unsaturated FA supplementation. By feeding unsaturated FA-enriched diets, the mammary gland reduced overall fat synthesis, leading to milk fat depression [37], but also carbohydrate synthesis, with a contemporary increase in carbohydrate catabolism (TCA cycle, glycolysis, pyruvate metabolism) (Appendix A).

All of these pathways also control cellular metabolism, growth and proliferation, cell migration, and apoptosis, and are fundamental to mammary gland lactation [32]. Little information is available on the role of unsaturated FA (UFA) supplementation on regulation of genes in modifying mammary glands of ewes at the structural and tissue level. Only a previous study suggested that inhibition of cell proliferation and remodelling occurring in response to UFA-enriched diet can promote a modification in milk synthesis in cows [13].

This complex adaptation was regulated by the transcription of the majority of candidate genes involved in insulin signaling, oxidative phosphorylation, PI3K-Akt signaling pathway and Wnt signaling pathway (Appendix A). Among these pathways, only oxidative phosphorylation showed significant enrichment (Figure 3).

One of the genes involved in these pathways is *AKT serine/threonine kinase 2* (*AKT2*), which was down-regulated with L diet. This gene is responsible for insulin signaling regulation [38] and encodes the AKT kinase, which regulates glucose uptake by mediating insulin-induced translocation of the SLC2A4/GLUT4 glucose transporter to the cell surface. A previous study suggested a total dependence of milk synthesis on adequate insulin level [39]. High fat diet was shown to alter the insulin level in the mammary gland, which was associated with the reduction of de novo fatty acids synthesis [39].

Insulin transduction is a biochemical pathway by which insulin increases the uptake of glucose into fat and muscle cells and reduces the synthesis of glucose in the liver and hence is involved in maintaining glucose homeostasis. This pathway is also influenced by fed versus fasting states, stress levels, and a variety of other hormones. Moreover, Menzies and co-workers [40] demonstrated that insulin regulates milk protein synthesis in the mammary gland. In this work, linseed supplementation induced a significant decrease in the expression of the genes of this pathway. This result may be related to the reduction of milk protein secretion observed in L ewes (Table 1). The elongation phase of protein synthesis in eukaryotes is stimulated by insulin through phosphorylation of eukaryotic elongation factor 1 (eEF1) [41] and dephosphorylation of eukaryotic elongation factor 2 (eEF2) [42]. The dephosphorylation of eEF2 results in increased eEF2 activity and rate of peptide elongation. In ewes fed L diet, we observed the down-regulation of transcripts encoding *eEF2K* (*eukaryotic elongation factor 2 kinase*), an enzyme that regulates protein synthesis by controlling the rate of peptide chain elongation, suggesting transcriptional regulation of the reduction in milk protein secretion. Moreover, the regulation of *EEF2K* was related to a series of genes involved in the insulin transduction pathway that were under-expressed after L diet: *GSK3B* and *SOS2* [43].

An important role attributed to dietary linseed supplementation is the anti-inflammatory process, played by n3 PUFA, particularly ALA. It has been demonstrated that this FA is an important regulator of anti-inflammatory substance release, such as cytokines [44]. Cytokines are immunoregulatory mediators that play a fundamental role in the regulation of immune responses against different infections, and also in the mammary gland defenses [44]. Two of the up-regulated genes of this pathway encode chemokines (*IL17RE* and *CXCR2*), which are generally associated with chemotaxis of leucocytes. Although the role of chemokines in the mammary gland remains unknown, L diet may contribute to the increase in the immune response of mammary glands, as shown by Caroprese et al. [44].

Moreover, we revealed the up-regulation of genes involved in the cytokine–cytokine receptor interaction, such as *TNFRSF6B, TNFRSF18, TNFRSF11A, TNFRSF13B, TNFSF13, C-C chemokine receptor type 3* (*CCR3*) signaling in eosinophils, *CXC chemokine receptor 2* (*CXCR2*) signaling and integrin signaling, which play a role in the differentiation and/or apoptosis in human mammary cells [45]. Cytokine signaling exerts potent chemokinetic and chemotactic activity on leukocytes and enhances the bactericidal activity of phagocytes in dairy cows [45]. Lessard et al. [45] suggested that cellular immunity of the dairy cows was affected by dietary supplementation of UFA. They observed that 5 days after calving, the lymphocyte proliferative response of cows allocated to linseed treatment was reduced. Though little is known about the expression of defense, inflammatory and immune-related genes in response to dietary UFA supplementation in dairy ewes, the results presented here suggest that enriched-UFA diets may affect immune functions of the mammary gland and thus may modify the susceptibility to mastitis in lactating sheep and the resulting quality of milk. Experiments specifically designed to test these hypotheses are needed to verify the roles of UFA on genes involved in immune system response pathways and networks, together with cell cycle, cell growth and certain apoptotic pathways [13].

An important pathway enriched with linseed supplementation is the P53 signaling pathway, which is involved in damage repair and cell proliferation [46]. Some of the up-regulated genes (*APAF3, CCNG1, CCNG2* and *PPM1D*) are intermediates for P53 activity, which is involved in cellular growth arrest, apoptosis, angiogenesis and DNA repair [47]. Our data demonstrated that linseed supplementation promotes mammalian anti-cancer activity as reported in previous studies [48].

L diet induced an over-expression of transcripts encoding two isoforms of casein kinase (*CSNK1A1* and *CSNK2B*). Casein kinases are operationally defined by their preferential utilization of acidic proteins such as caseins as substrates. However, these molecules can phosphorylate many proteins such as CTNNB1, PER1 and PER2. These proteins are known to be involved in regulation of circadian clocks in many organs [49], so may play a role in keratin cytoskeleton disassembly and, thereby, may regulate epithelial cell migration and cell adhesion [40]. Moreover, these proteins also participate in Wnt signaling [49]. Wnt is a secreted glycoprotein, which is involved in autocrine or paracrine activity [50]. It has an important role in the regulation of cell proliferation, differentiation, and migration during an organism’s growth and development and may have a role in segregating chromosomes during mitosis [50]. Recent studies have shown that the Wnt signaling pathway is necessary to regulate the normal development of the mammalian reproductive system [50]. The role of up-regulation of casein kinase genes in mammary gland of L ewes may be related to epithelial cell adhesion and should be further investigated.

Another pathway that showed enhanced activity after L diet was oxidative phosphorylation, involved in the release of energy to produce ATP, by a redox reaction that involves a series of protein complexes, within the inner membrane of the cell’s mitochondria. In this pathway, 21 genes showed an up-regulation, while only two were down-regulated after L diet. Several of the up-regulated genes are related to two of the five membrane complexes involved in the membrane respiratory chain: Complex I (*NADH dehydrogenase*) and Complex V (*ATP synthase*). Both complexes showed a higher activity in L sheep, demonstrating an enhanced regulation of the oxidative pathway: Complex I regulates the transfer of electrons from NADH to the respiratory chain by ubiquinone, while Complex V produces ATP from ADP in the presence of a proton gradient across the membrane, which is generated by electron transport complexes of the respiratory chain [51]. In particular, the genes of Complex I encoding 3 core subunits (*NDUFS7*, *NDUFS8* and *NDUFV1*) and 7 accessory subunits (*NDUFA3, NDUFA7, NDUFA10, NDUFA13, NDUFB7, NDUFB10, NDUFB11*) of NADH dehydrogenase and all genes encoding ATP synthase subunits of the Complex V were up-regulated. Besides these complexes, two genes involved in oxidative metabolism were also differentially expressed after L diet supplementation: *PPA2* (encoding inorganic pyrophosphatase 2) and *LHPP* (encoding phospholysine phosphohistidine, an inorganic pyrophosphate phosphatase), which catalyze the hydrolysis of inorganic pyrophosphate and imidodiphosphate respectively, and are essential for maintaining mitochondrial membrane potential [51]. These results suggest metabolic changes in mammary cells of L ewes are energy consuming and highly regulated at the transcriptional level.

### 4.2. Milk Production and Composition

The results of our study showed that linseed supplementation reduces the milk yield by 30%. This data is not in agreement with the studies of Zhang et al. [52] and Gómez Cortés et al. [53], who reported that supplementing basal diet with extruded linseed, resulted in a milk yield increase compared with the control treatment. Generally, supplemental fat is fed to increase the energy density of the diet and may result in increased milk production. However, milk production does not always respond positively to fat supplementation. In fact, it was observed that fat supplementation increased [54], had no effect [55], or decreased [56] milk yield of dairy ruminants. The response of dairy animals to fat supplementation is affected by several factors, as proposed by Zhang et al. [52].

In our work, the decrease in milk production may be related to a reduction in lactose synthesis, which controls milk volume by maintaining the osmolarity of milk [57]. Indeed, we found a lower level of lactose (−30%) in milk of L ewes and this reduction may be related to a downregulation of *beta-1,4-galactosyltransferase 1* (*B4GALT1*) as confirmed by the positive correlation coefficient (r = 0.78, *p* < 0.01) (Appendix A), which interacts with alfa-lactalbumin to form the lactose synthase complex that produces lactose (Appendix A).

Biochemical data indicate that the L diet decreased the secretion of fat (−25%) and protein (−30%), in agreement with a previous work [52]. The reduction of fat is confirmed by the observed significant decrease in the concentrations of principal SFA measured, which derived principally by mammary de novo synthesis [1]. For example, the concentration of C12:0, C14:0 and C16:0, decreased by 54%, 32%, 21% respectively at the end of supplementation. These data confirmed the efficacy of linseed to increase the concentrations of healthy FA in milk while decreasing SFA. In particular, L diet increased the level of PUFA, especially the omega-3 fraction (+33%) [9].

However, we observed an absence of effects due to L diet on the mRNA abundance of the four principal lipogenic enzymes (*ACACA*, *FASN*, *DGAT1*, *SREBP1*), despite the decrease in the secretion of medium chain FA (C12:0–C16:0) and the simultaneous increase in C18-FA secretion.

This lack of effect on the expression of a few candidate genes has been reported previously in goats receiving lipid supplements [7,58] and is in contrast with results obtained in ewes fed with fish oil or trans-10 cis-12 CLA [14,15]. L diet probably reduced lipid secretion by mechanisms other than reducing ACACA and FASN mRNA abundances, as suggested by Faulconnier et al. [58]. For example, the downregulation of *DGKE*, involved in glycerophospholipid metabolism (Appendix A), is known to affect lipogenic genes [59]. The *DGKE* gene encodes a cytoplasmic enzyme that phosphorylates diacylglycerol to produce phosphatidic acid, which is also linked to the regulation of diverse functions, including cell growth and lipid metabolism [60].

On the contrary, the *SCD* transcript level showed a significant decrease, which is in accordance with previous studies investigating diets supplemented with fish oil [61]. This decrease in *SCD* mRNA abundance is in line with the reduction of all the desaturase index (DI) ratios, principally DI14 (C14:1c9/(C14:1c9 + C14:0)) (Table 1), which is considered the best proxy for Δ-9 desaturation activity in mammary tissues [7]. A high correlation (r = 0.81, *p* < 0.01) was observed between *SCD* expression and DI14, demonstrating a strict relationship between gene expression and enzyme activity.

Regarding milk PUFA in L ewes, we observed a significant increase in PUFA related to diet or ruminal activity (CLA, C18:2 c9t11) and a reduction in very long chain PUFAs like eicosapentaenoic (C20:5n3) and docosahexaenoic acid (C22:6n3) (Table 1). Synthesis of these PUFAs is carried out by *fatty acid desaturase 1* (*FADS1*) and *2* (*FADS2*), which add double bonds at the Δ5 and Δ6 position, respectively [9], so the reduction might be related to a lower expression of these genes. However, in our study no differential expression was observed for these genes, though the level of *FADS1* transcription showed a non-significant reduction in the mammary gland of L ewes fed with linseed (Appendix A). The control of FADS activity is probably related to the protein level. On the contrary, we observed a down-regulation of *ELOVL5* and *ELOVL6* genes, which encode the enzymes responsible for the long-chain elongation cycle, possibly explaining their reduction.

In our study, the concentrations of C18:1t11 and C18:2c9t11 were significantly increased by L treatments. As known, extruded linseed affected the ruminal biohydrogenation process and related pathways differently leading to ruminal outflow FAs that were absorbed and incorporated into milk. Diet composition affects ruminal biohydrogenation pathways producing a wide range of positional and geometric isomers and modified FAs including non-conjugated and partially conjugated C18:2 and C18:3 isomers [62]. In our study, the concentrations of C18:1t11 and C18:2c9t11 were significantly increased by L treatments. A high proportion of milk C18:2c9t11 is synthesized endogenously in the mammary gland using ruminal C18:1t11 as a substrate [5].

Hence our data suggest that milk PUFA content during supplemental feeding with linseed was related to an increased availability of ALA and biohydrogenation metabolites (mostly PUFAs) that were taken up and incorporated into milk, but also by gene regulation as suggested by transcriptomic analyses.

Genes significantly regulated by L diet like *ANGPTL4* (*angiopoietin-like 4*) could be important candidates in pooled pathways of mammary lipogenesis/FA uptake. The liver and adipose tissues have been identified as key sources of ANGPTL4, an adipokine involved in the regulation of lipid metabolism in cattle [63]. It was demonstrated that its expression changes with altered energy balance in lactating dairy cattle [64]. The up-regulation of *ANGPTL4* is predicted to increase the concentration of cholesterol and the quantity of steroids. Moreover, it was shown in transgenic mice that *ANGPTL3* and *ANGPTL4* regulated circulating triglyceride levels under different nutritional regimes and thus play roles in lipid metabolism through differential inhibition of lipoprotein lipase [65]. This effect should be related to a contemporary increase in transcripts of genes involved in the triacylglycerol synthesis, such as *GPAT4, AGPAT1, AGPAT2* and *AGPAT3*. *GPAT4* catalyzes the first committed step in glycerolipid biosynthesis, the acylation of glycerol-3-phosphate, with the synthesis of 1-acyl-sn-glycerol-3-phosphate. This reaction represents the essential step in the glycerolipid synthesis. The three AGPAT genes encode acyl-CoA:1-acylglycerol-sn-3-phosphate acyltransferase, which catalyzes the second acylation in the sn2 position [66]. On the contrary, no effect was revealed for the DGAT1 gene, which catalyzes the last step of triacylglycerol synthesis. Moreover, we observed a down-regulation of *LPIN3*, encoding for the magnesium-dependent phosphatidate-phosphatase enzyme, which catalyzes the conversion of phosphatidic acid to diacylglycerol during triglyceride, phosphatidylcholine and phosphatidylethanolamine biosynthesis. Our data confirmed the role of changes in energy status of the animal on lipid metabolism, as a consequence of dietary modifications. This adaptation is possibly mediated by modulation of *ANGPTL4* expression and related genes [64].

Cholesterogenic activity was also affected by linseed supplementation, by the down-regulation of several genes (*LSS*, *DHCR24*, *FDFT1*, *HSD17B7*, *MSMO1*, *SC5D*, *SQLE*) (Appendix A) that are targets of transcription factor family SREBFs. This regulation does not seem related to changes in expression level of SREBF transcripts that remained unchanged in our experiment, suggesting a post-transcriptional rather than transcriptional regulation Similar results were observed in ewes fed with fish oil [14] where genes included in SREBP signaling pathway were downregulated but SREBF1 gene was not differentially expressed with respect to control.

Our data provide information into the nutrient metabolism adaptations in the mammary gland after unsaturated FA supplementation. We suggest that, through feeding linseed-enriched diets, the mammary gland reduced overall fat and protein metabolic activity, but increased carbohydrate metabolism. As a matter of fact, most of the transcripts involved in the biological process related to carbohydrate metabolism (glycolysis and gluconeogenesis, and pentose phosphate pathway) were upregulated.

## 5. Conclusions

The results of our study propose that supplementing the diets of dairy ewes with linseed UFAs decreases milk fat, protein and lactose secretion, and reduces milk yield. As a consequence of ALA supplementation, the proportion of PUFA n3 and CLA in milk increases, whereas de novo FA synthesis decreases, with an improvement in the nutrition quality of ewe milk. The linseed supplementation determined transcriptional adjustments of 3928 genes, suggesting a strong impact on metabolism and other cellular functions in the mammary gland. The functional analysis of these genes indicated that inclusion of dietary UFAs modifies the expression of genes involved in lipid and protein metabolism. Moreover, linseed supplementation also modified the regulation of genes involved in cell–cell interactions, cells morphology (cytoskeleton organization), cell death and the immune response. These transcriptional adaptations occurring in mammary tissue in response to dietary lipids might provide new perspectives for more detailed functional studies of the mammary metabolism of ewes.

## Figures and Tables

**Figure 1 animals-11-02707-f001:**
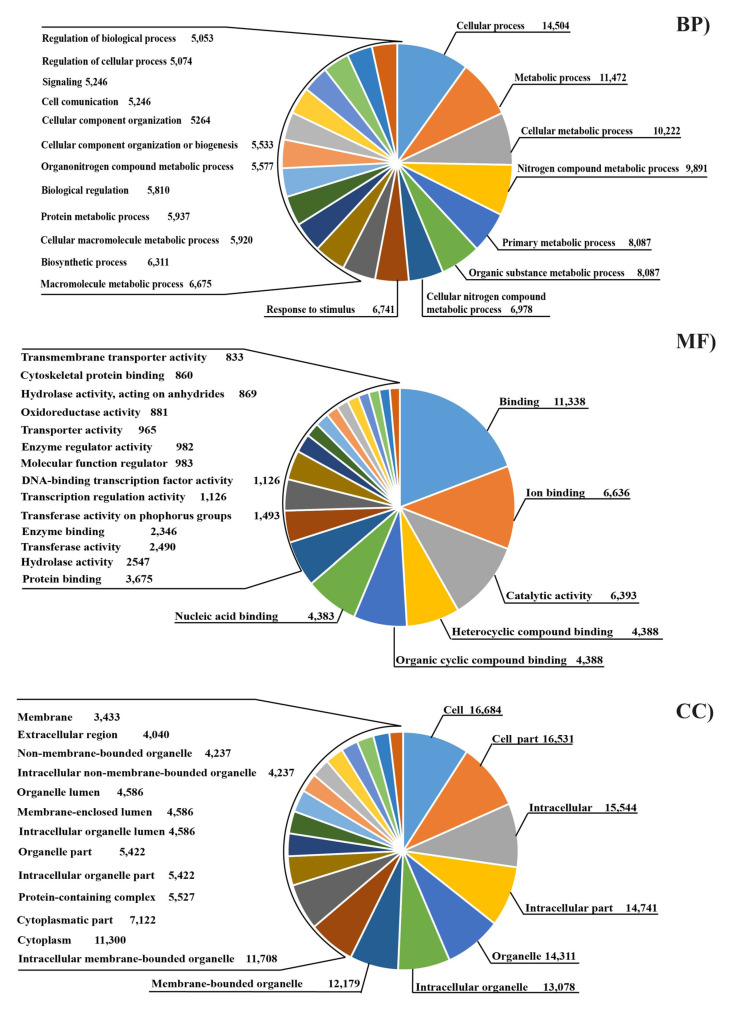
Gene ontology terms distribution for whole expressed genes (RPKM > 1 in at least one library) for control and linseed fed sheep. Cut off was set for the top 20 GO terms. Numbers indicate genes belonging to each GO term.

**Figure 2 animals-11-02707-f002:**
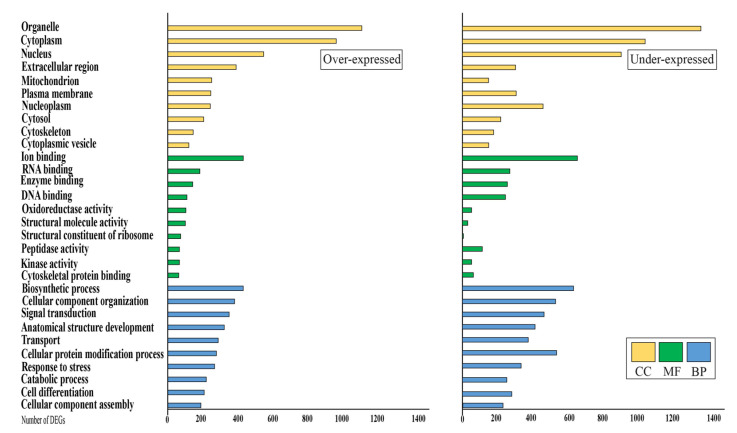
Distribution of gene ontology terms for over and under differentially expressed genes in sheep fed with linseed. Threshold was set to the top 10 most represented terms per major gene ontology classes. BP = biological process; MF = molecular function; CC = cellular component; DEGs = differentially expressed genes.

**Figure 3 animals-11-02707-f003:**
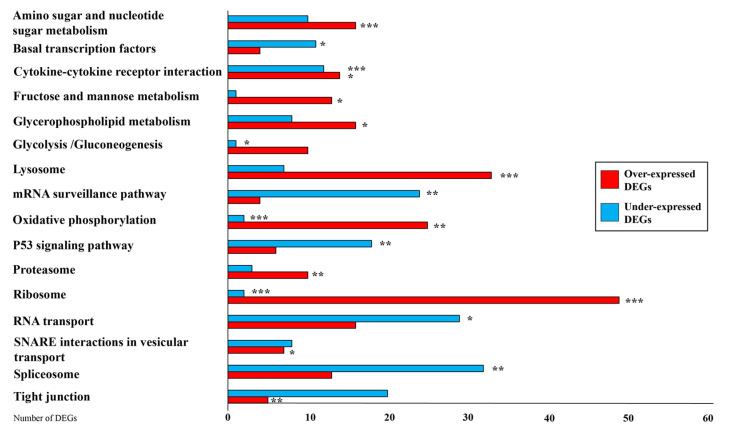
Differentially expressed gene distribution for enriched KEGG. Red bars represent over expressed genes and blue bars stand for under expressed transcripts. * = *p*-Value ≤ 0.05; ** = *p*-Value ≤ 0.01; *** = *p*-Value ≤ 0.001; DEGs = differentially expressed genes.

**Table 1 animals-11-02707-t001:** Effect of linseed supplementation on milk production and composition and fatty acid profile.

		C ^1^	L ^2^	SE ^3^	*p*-Value
		*n* = 8	*n* = 8		
Milk production	g/d	1148.36	798.55	68.91	<0.001
Fat	g/100 g of milk	7.35	7.85	0.26	0.211
Lactose	g/100 g of milk	4.34	4.40	0.06	0.424
Proteins	g/100 g of milk	6.51	6.49	0.11	0.942
Fat secretion	mg	83.99	62.70	5.85	0.004
Lactose secretion	mg	50.51	35.39	3.35	0.003
Protein secretion	mg	73.88	51.22	4.07	<0.001
C4:0	g/100 g of total lipids	3.11	2.62	0.07	<0.001
C6:0	g/100 g of total lipids	2.52	1.45	0.06	<0.001
C8:0	g/100 g of total lipids	2.41	1.16	0.08	<0.001
C10:0	g/100 g of total lipids	7.43	3.23	0.27	<0.001
C10:1c9	g/100 g of total lipids	0.29	0.11	0.01	<0.001
C11:0	g/100 g of total lipids	0.07	0.03	0.01	<0.001
C12:0	g/100 g of total lipids	4.28	1.96	0.16	<0.001
C13:0iso	g/100 g of total lipids	0.03	0.01	0.00	<0.001
C13:0ante	g/100 g of total lipids	0.01	0.01	0.00	0.202
C12:1c11	g/100 g of total lipids	0.06	0.02	0.00	<0.001
C13:0	g/100 g of total lipids	0.06	0.04	0.00	<0.001
C14:0iso	g/100 g of total lipids	0.12	0.07	0.00	<0.001
C14:0	g/100 g of total lipids	8.63	5.89	0.24	<0.001
C15:0iso	g/100 g of total lipids	0.27	0.14	0.01	<0.001
C15:0ante	g/100 g of total lipids	0.44	0.27	0.01	<0.001
C14:1c9	g/100 g of total lipids	0.17	0.10	0.01	<0.001
C15:0	g/100 g of total lipids	0.94	0.71	0.02	<0.001
C16:0iso	g/100 g of total lipids	0.27	0.14	0.01	<0.001
C16:0	g/100 g of total lipids	19.93	15.62	0.54	<0.001
C16:1t6/7	g/100 g of total lipids	0.02	0.06	0.00	<0.001
C16:1t9	g/100 g of total lipids	0.04	0.32	0.01	<0.001
C17:0iso	g/100 g of total lipids	0.37	0.26	0.01	<0.001
C16:1c7	g/100 g of total lipids	0.23	0.26	0.01	0.035
C16:1c9	g/100 g of total lipids	0.74	0.52	0.03	<0.001
C17:0ante	g/100 g of total lipids	0.42	0.22	0.01	<0.001
C16:1c13	g/100 g of total lipids	0.13	0.04	0.01	<0.001
C17:0	g/100 g of total lipids	0.59	0.43	0.02	<0.001
C17:1c9	g/100 g of total lipids	0.24	0.12	0.01	<0.001
C18:0	g/100 g of total lipids	5.20	8.87	0.35	<0.001
C18:1t6-8	g/100 g of total lipids	0.17	0.96	0.03	<0.001
C18:1t9	g/100 g of total lipids	0.15	0.72	0.02	<0.001
C18:1t10	g/100 g of total lipids	0.21	0.81	0.03	<0.001
C18:1t11	g/100 g of total lipids	0.67	4.94	0.17	<0.001
C18:1t12	g/100 g of total lipids	0.24	0.83	0.02	<0.001
C18:1c9	g/100 g of total lipids	15.39	21.81	0.55	<0.001
C18:1t15	g/100 g of total lipids	0.15	0.46	0.02	<0.001
C18:1c11	g/100 g of total lipids	0.34	0.45	0.01	<0.001
C18:1c12	g/100 g of total lipids	0.23	0.29	0.01	<0.001
C18:1t16	g/100 g of total lipids	0.19	0.51	0.02	<0.001
C18:1c14	g/100 g of total lipids	0.04	0.09	0.00	<0.001
C18:2n6	g/100 g of total lipids	3.03	2.06	0.11	<0.001
C18:3n3	g/100 g of total lipids	1.41	2.18	0.10	<0.001
C18:2c9t11	g/100 g of total lipids	0.47	2.38	0.08	<0.001
C22:0	g/100 g of total lipids	0.16	0.15	0.01	0.196
C20:3n6	g/100 g of total lipids	0.03	0.01	0.00	<0.001
C20:3n3	g/100 g of total lipids	0.03	0.02	0.00	0.043
C20:4n6	g/100 g of total lipids	0.22	0.10	0.01	<0.001
C23:0	g/100 g of total lipids	0.12	0.09	0.01	0.011
C20:5n3	g/100 g of total lipids	0.09	0.06	0.00	<0.001
C24:0	g/100 g of total lipids	0.08	0.07	0.00	0.832
C22:3n3	g/100 g of total lipids	0.02	0.01	0.00	0.003
C22:5n3	g/100 g of total lipids	0.19	0.09	0.00	<0.001
C22:6n3	g/100 g of total lipids	0.07	0.04	0.00	<0.001
SFA	g/100 g of total lipids	57.45	43.43	1.02	<0.001
MUFA	g/100 g of total lipids	19.67	33.43	0.74	<0.001
PUFA	g/100 g of total lipids	5.60	6.97	0.23	<0.001
PUFAn6	g/100 g of total lipids	3.28	2.12	0.11	<0.001
PUFAn3	g/100 g of total lipids	1.84	2.45	0.10	<0.001
BCFA	g/100 g of total lipids	1.92	1.12	0.04	<0.001
OCFA	g/100 g of total lipids	3.19	2.08	0.07	<0.001
n6/n3		1.87	0.87	0.07	<0.001
DI10 ^4^		0.04	0.03	0.00	<0.001
DI14 ^4^		0.02	0.02	0.00	0.016
DI18 ^4^		0.75	0.71	0.01	0.013
DI/RA ^4^		0.43	0.33	0.01	<0.001

^1^ Control diet; ^2^ Linseed diet; ^3^ Standard Error; ^4^ Desaturase Index.

**Table 2 animals-11-02707-t002:** Summary statistics for the Illumina sequencing and aligning against *Ovis aries* reference transcriptome Oar_rambouillet_v1.0. C = control sheep; L = linseed fed sheep.

Library	Number of Raw Reads	Number of Trimmed Reads	Number of Mapped Reads on *Ovis aries* Reference Transcriptome	Percentage of Mapped Reads on *Ovis aries* Reference Transcriptome
C1	28,098,370	25,682,138	24,120,413	93.92%
C2	26,880,020	23,336,744	20,150,560	86.35%
C3	48,397,516	39,530,734	37,211,397	94.13%
C4	25,537,932	20,359,560	16,085,295	79.01%
L1	45,305,912	23,821,170	20,585,520	86.42%
L2	34,764,246	11,941,984	10,888,233	91.18%
L3	26,732,292	24,451,846	22,165,383	90.65%
L4	46,172,488	18,620,748	16,039,894	86.14%

## Data Availability

The datasets used and/or analyzed during the current study are available from the corresponding author on reasonable request.

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
