# Peer review of "Transcriptome Adaptation of the Ovine Mammary Gland to Dietary Supplementation of Extruded Linseed"

_animals, 2021, doi:10.3390/ani11092707_

Round 1
Reviewer 1 Report
In the completed and revised version of the article, the authors have expanded and supplemented in the discussion the information regarding the length of the milk sampling period and the performance of the sheep during this period. This information in addition to the given literature is sufficient. The literature cited in the article has been systematized. I have no comments on the rest of the article. I believe that the article in this form meets the requirements of the Animals Journal.
Reviewer 2 Report
In this paper, the authors performed a RNA sequencing analysis on the ovine mammary gland following supplemental feeding with 5% linseed panel. The overall design is reasonable, and interesting. However, the manuscript lacks some details. Major revision is required before any further decisions.
- A major highlight of this studylies in the detection of milk composition and fatty acid profile of all samples. However, the author did not perform any correlation or interaction analysis of phenotypic data and transcriptome data, which undoubtedly greatly weakened the significance of the analysis results.
- The author may need to carry out QPCR confirmation of the identified differential genes in RNA-Seq.
- Why add 5% linseed in the feed, please explain。
- It appears that 8 ewes were applied in the per treatment, and four samples each group were selected for sequencing. Why 4? What are the selection criteria? And how many samples were used for phenotyping?
- Line240, please provide the SRA accession codes of all the sequences.
- It is recommended that the threshold of functional enrichment analysis could be set to FDR<0.01 or FDR<0.05.
- I suggested providing the correlation among the sequencing samples in the result section.
- Line 236 and Table 2, provide the correct genome version.
- Please use italics for gene names in the manuscript.
- Please provide Figure S1.
Round 2
Reviewer 2 Report
The author makes a revision to the relevant problems. This paper still has defects.
In the section of discussion, no found the correlation among the sequencing samples.
The thresholds for the material method(Line 187)and results sections (Line 241) are different.
please provide proof that the data has been deposited in the form of a confirmation email from the repository.
It is recommended that the threshold of functional enrichment analysis could be set to FDR<0.01 or FDR<0.05 (Lines 196-197).
I suggested that the revised part be marked in the manuscript.
Round 3
Reviewer 2 Report
Thanks for the authors' cooperation, the concerns I listed have been addressed appropriately.
This manuscript is a resubmission of an earlier submission. The following is a list of the peer review reports and author responses from that submission.
Round 1
Reviewer 1 Report
The manuscript of Conte and colleagues presents RNA seq data obtained from milk somatic cells following treatment of ewes with extruded linseed. While this is a potentially interesting study unfortunately it lacks appropriate control as is explained below. Therefore, the effects observed are impossible to attribute to the treatment and thus the conclusions drawn cannot be supported by the results.
While it is an interesting approach to use the same animals before and after treatment, here the difference in the stage of lactation between control and treatment groups introduces a confounding effect of the lactation stage that in sheep is expected to significantly affect both daily milk yield and content. In addition, milk transcriptome studies have reported differential gene expression between lactation stages (e.g., Suárez-Vega et al. BMC Genomics, 2017, 18:170). Thus, this experimental design is not appropriate to reliably observe the effects of linseed dietary treatment as it does not allow to distinguish between the effects of linseed supplementation from the effect of the difference in lactation stage on either milk yield and quality parameters or gene expression.
In the present study, data for the control group are collected from ewes at 117+/-12 days in lactation, if I calculate correctly from the methods, while the samples for the linseed treatment group are collected from the same ewes at 147+/-12 days in lactation. The authors argue that a month is not enough time to make a significant difference in milk production and base this argument on data from dairy cows. In ewes lactation curves differ significantly from cattle and in most breeds daily milk yield declines quickly following peak. The authors should therefore provide data for the lactation curve of the Comisana breed to support the claim that 31 days do not have a significant impact on daily milk yield and milk content as well as gene expression in the mammary gland due to lactation stage differences. Contrary to the authors’ argument, and in line with most sheep breeds, published data for the lactation curve of the Comisana breed that I was able to retrieve (e.g. Portolano et al. Small ruminant research, 1996, 24:7-13) show that there is no period with constant daily milk yield and a continuous decline in daily milk yield after peak lactation is observed. In another report for the breed, the period 100-130 days in lactation is characterized as mid lactation while 150-180 days late lactation, thus control and treated samples are collected at different stages of lactation (Sevi et al. Small Ruminant Research, 2004, 51: 251-259).
Reviewer 2 Report
Human and animal nutrition has a direct effect on the body and consequently on gene expression and physiological response to the diet used. In the study presented here, the authors conducted nutritional studies on lactating Comisana mothers in mid-lactation, receiving in ration a standard mixture in the control group and a control mixture with the addition of linseed. After the 28th feeding period, milk was collected and analyzed for chemical composition as well as gene expression. In the second part of the experiments, the experimental groups were switched and the same feeds were fed and identical determinations were made after 28 days. This treatment, strengthens the statistical power using only 8 mother sheep at a time.
Specific comments:
The article is written in very good scientific language. The research and especially analytical methods used were appropriate to the type of analyses performed. I have a comment in the form of a discussion whether it is not necessary before studying the effects of diet on gene expression not to test and standardize the animals to be "genetically equal". A model of such genetically standardized animals would be a great model for human research. I have no comments on the remaining sections of the article, i. e. the results and discussion I think the paper is well written and with minor additions can be published in Animals.